# Design of multi-scale protein complexes by hierarchical building block fusion

Yang Hsia [1,2,3,10], Rubul Mout [1,2,10], William Sheffler[1,2], Natasha I. Edman [1,2,4,5], Ivan Vulovic[1,2,6], Young-Jun Park [1], Rachel L. Redler[7], Matthew J. Bick[1,2], Asim K. Bera[1,2], Alexis Courbet[1,2], Alex Kang[1,2], T. J. Brunette[1,2], Una Nattermann[1,2,3], Evelyn Tsai[1,2], Ayesha Saleem[1,2], Cameron M. Chow [1,2], Damian Ekiert [7,8], Gira Bhabha[7], David Veesler [1] & David Baker [1,2,9✉]

A systematic and robust approach to generating complex protein nanomaterials would have broad utility. We develop a hierarchical approach to designing multi-component protein assemblies from two classes of modular building blocks: designed helical repeat proteins (DHRs) and helical bundle oligomers (HBs). We first rigidly fuse DHRs to HBs to generate a large library of oligomeric building blocks. We then generate assemblies with cyclic, dihedral, and point group symmetries from these building blocks using architecture guided rigid helical fusion with new software named WORMS. X-ray crystallography and cryo-electron micro-scopy characterization show that the hierarchical design approach can accurately generate a wide range of assemblies, including a 43 nm diameter icosahedral nanocage. The computational methods and building block sets described here provide a very general route to *de novo* designed protein nanomaterials.

[1] Department of Biochemistry, University of Washington, Seattle, WA, USA. [2] Institute for Protein Design, University of Washington, Seattle, WA, USA. [3] Biological Physics, Structure and Design Graduate Program, University of Washington, Seattle, WA, USA. [4] Molecular and Cellular Biology Graduate Program, University of Washington, Seattle, WA, USA. [5] Medical Scientist Training Program, University of Washington, Seattle, WA, USA. [6] Molecular Engineering Graduate Program, University of Washington, Seattle, WA, USA. [7] Department of Cell Biology and Skirball Institute of Biomolecular Medicine, New York University School of Medicine, New York, NY, USA. [8] Department of Microbiology, New York University School of Medicine, New York, NY, USA. [9] Howard Hughes Medical Institute, University of Washington, Seattle, WA, USA. [10] These authors contributed equally: Yang Hsia, Rubul Mout. ✉email: dabaker@uw.edu

There has been considerable recent progress in designing self-assembling protein nano structures and materials mediated by non-covalent protein–protein interactions, metal-mediated interactions, or peptide linkers[1–5]. Through the design of non-covalent protein–protein interfaces, proteins have been created that self-assemble into a wide variety of higher-order structures, from cyclic[6] and dihedral symmetries[7] to point group nanocages[8–10], 1-dimensional fibers[11], and two-dimensional arrays[12]. However, non-covalent protein interface design remains challenging, and interface quality is dependent on how well the building blocks complement each other during design. A second approach has exploited metal coordination chemistry to design metal-mediated protein–protein interfaces in a variety of different symmetries; these materials can have quite unusual mechanical properties[4].

A third approach that avoids the need for designing new interfaces is to fuse oligomeric protein building blocks with helical linkers. This approach has generated a number of new materials[13], but in many cases the lack of rigidity has made the structures of these assemblies difficult to precisely predict. More rigid junctions created by overlapping ideal helices and designing around the junction region have yielded more predictable structures[14,15], including closed ring dihedral structures which require even more precise structure predictions[16]. This rigid fusion method, however, has its own set of challenges in comparison to designing a new non-covalent protein–protein interface: first, for any pair of protein building blocks, there are far fewer positions for rigid fusion than are for unconstrained protein–protein docking limiting the space of possible solutions, and second, while in the non-covalent protein interface case space searched can be limited by restricting building blocks to the symmetry axes of the desired nanomaterial, this is not possible in the case of rigid fusions, making the search more difficult as the number of building blocks increases.

A potential solution to the issue of having smaller numbers of possible fusion positions for a given pair of building blocks in the rigid helix fusion method is to systematically generate large numbers of building blocks having properties ideal for helix fusion. Attractive candidates for such an approach are *de novo* helical repeat proteins (DHRs)[17] consisting of a tandemly repeated structural unit, which provide a wide range of struts of different shape and curvature for building nanomaterials, and parametric helical bundles (HBs)[18–21] which provide a wide range of preformed protein–protein interfaces for locking together different protein subunits in a designed nanomaterial. Many examples of both classes of designed proteins have been solved by X-ray crystallography, and they are typically very stable. We reasoned that by systematically fusing DHR "arms" to central HB "hubs" we could generate building blocks with a wide range of geometries and valencies that, because of the modular nature of repeat proteins, enable a very large number of rigid helix fusions: given two such building blocks with N- and C-terminally extending repeat protein arms, the potentially rigid fusion sites are any pair of internal helical residues in the DHR arms. With a large library of building blocks, the challenge is then to develop a method to very quickly traverse all possible combinations of fusion locations. Here we describe the use of geometric hashing of transforms to quickly and systematically identify the fusion positions in large sets of building blocks that generate any specified symmetric architecture, and the use of this approach to design a broad range of symmetric assemblies.

## Results
We describe the development of methods for creating large and modular libraries of building blocks by fusing DHRs to HBs, and then using them to generate symmetric assemblies by rapidly scanning through the combinatorially large number of possible rigid helix fusions for those generating the desired architecture. We present the methodology and results in two sections. In section one, we describe the systematic generation of homo- and hetero-oligomeric building blocks from *de novo* designed HBs, helical oligomers, and repeat proteins (Fig. 1a). In the second section, we describe the use of these building blocks to assemble a wide variety of higher-order symmetric architectures (Fig. 1c).

**Building block generation by rigid helical fusion of DHR arms to HB oligomers.** To generate a wide variety of building blocks, we explored two different methodologies for fusing DHRs to HBs (Fig. 1a). The first is to dock the DHR units to the HBs, redesign the residues at the newly created interface, and then build loops between nearby termini (HelixDock, HD). The second protocol simplifies the process by overlapping the helical termini of the DHRs and HBs and designing only the immediate residues around the junction (HelixFuse, HF). As an example of the combinatorial diversity that can be generated due to a large number of possible internal helical fusion sites in a DHR (nearly all helical residues), a single terminus from a single HB (2L6HC3-12[19], N-terminus) combined with the library of 44 verified DHRs results in 259 different structures (selected examples in Fig. 1b).

For the first approach, HelixDock (HD), 44 DHRs with validated structures[22] and 11 HBs[17,19] (including some without pre-verified structures) were selected as input scaffolds for symmetrical docking using a modified version of the sicdock software[6]. In each case, N copies of the DHR, one for each monomer in the HB, were symmetrically docked onto the HB, sampling all six degrees of freedom, to generate star-shaped structures with repeat protein arms emanating symmetrically from the HB in the center. Docked configurations with linkable N- and C-termini within a distance cutoff of 9 Å with interfaces predicted to yield low energy designs[6] were then subjected to Rosetta sequence design to optimize the residue identity and packing at the newly formed interface. Designs with high predicted domain-domain binding energy and shape complementarity[23] were identified, and loops connecting chain the termini were built using the ConnectChainsMover[15]. Structures with good loop geometry (passing worse9merFilter and FoldabilityFilter) were forward folded with RosettaRemodel[24] symmetrically, and those with sequences that fold into the designed structure *in silico* were identified.

Synthetic genes encoding a subset of the selected designs with a wide range of shapes were synthesized and the proteins expressed in *E. coli*. Of the 115 synthetic genes synthesized, 65 produced soluble protein. Of these, 39 had relatively monodisperse size exclusion chromatography (SEC) profiles that matched what was expected from the design, and were examined by small-angle X-ray scattering (SAXS), 17 had profiles close to those computed from the design models (Supplementary Figs. 1–3). Design C3_HD-1069, was crystallized and solved to 2.4 Å (Fig. 2a). Although the two loops connecting to the HB are unresolved in the structure, the resulting placement of the DHR remains correct (unresolved loops were also present in the original HB structure (2L6HC3_6)[19]. The resolved rotamers at the newly designed interface between the HB and DHR are also as designed.

For the second approach, HelixFuse (HF), the same set of DHRs and HBs were combinatorially fused together by overlapping the terminal helix residues in both directions ("AB": c-terminus of HB to n-terminus of DHR, "BA": n-terminus of HB to c-terminus of DHR)[15]. On the HB end, up to 4 residues were allowed to be deleted to maximize the sampling space of the fusion while maintaining the structural integrity of the oligomeric interface.

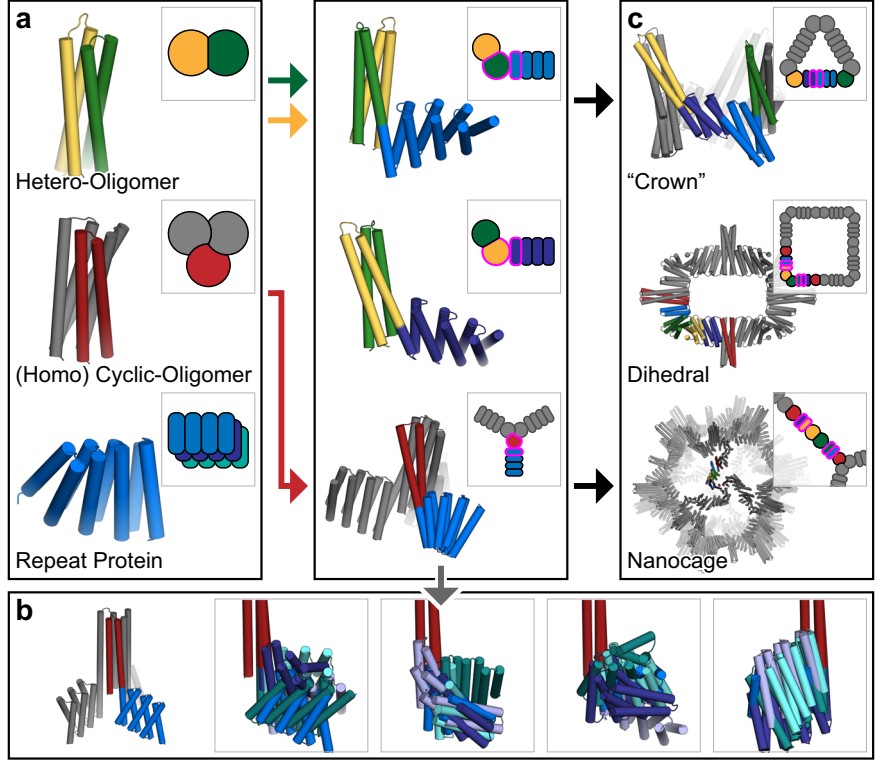

**Fig. 1 Overview of the rigid hierarchical fusion approach. a** Hetero- (yellow/green) and homo- (red) oligomeric helical bundles are fused to *de novo* helical repeat proteins (shades of blue) (left) to create a wide range of building blocks using HelixDock and HelixFuse (center). Symmetric units shown in gray. **b** Twenty representative HelixFuse outputs overlaid in groups of five display the wide range of diversity that can be generated by using a single helical bundle core (symmetric units hidden for clarity). **c** Building blocks are further assembled into higher-ordered structures through helical fusion (WORMS). The examples are cyclic crowns (top), dihedral rings (middle), and icosahedral nanocages (bottom); additional details available in their respective sections.

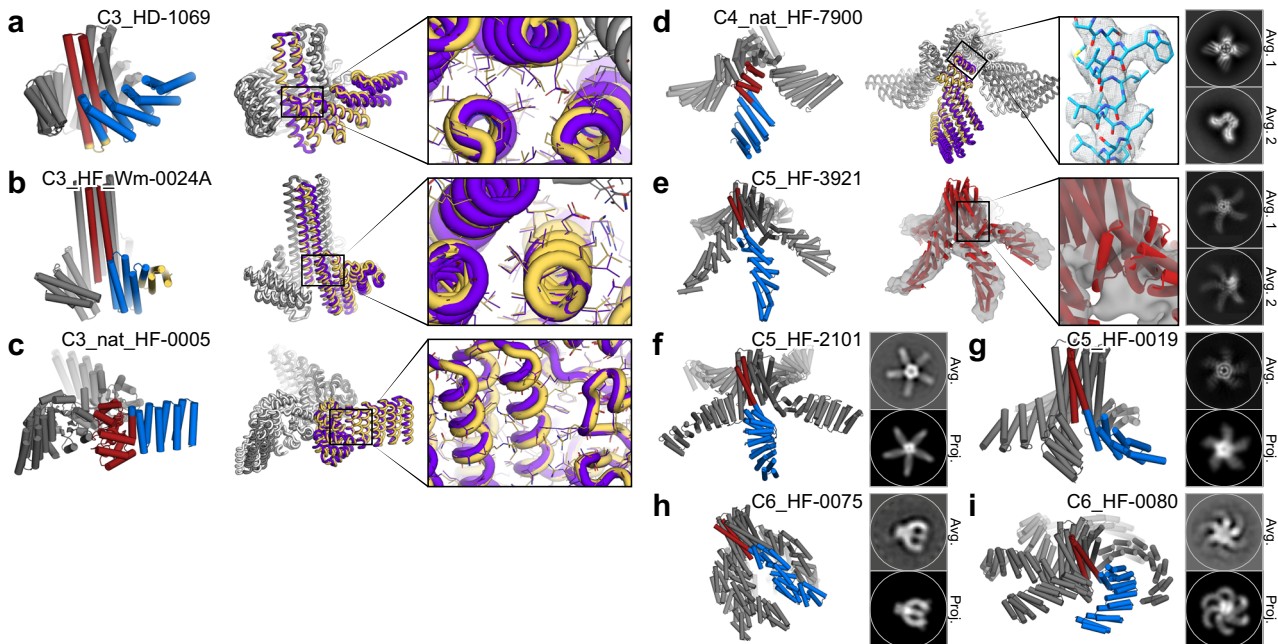

**Fig. 2 Homo-oligomer diversification by repeat protein fusion.** One central oligomer unit is shown in red and fused DHR in blue while other symmetrical units in grey. HelixDock (HD); HelixFuse (HF). **a** C3_HD-1069, with designed loop shown in yellow, **b** C3_HF_Wm-0024A, with additional WORMS fusion shown in yellow), and **c** C3_nat_HF-0005. Overlay of the design model (purple/gray) and crystal structure (yellow/white) shows the overall match of the backbone. Inset shows the correct placement of the rotamers in the designed junction region. **d** C4_nat_HF-7900; design model (purple/grey) and cryo-EM map (yellow/white), with insert highlighting the the high resolution (~3.8 Å) density. **e** C5_HF-3921 as characterized by cryo-EM, with inset showing density surrounding the designed junction. **f** C5_HF-2101, **g** C5_HF-0019, **h** C6_HF-0075, and **i** C6_HF-0080 all showed a good overall match to its negative-stain EM 2D class averages (top) from one direction, using a predicted projection map (bottom).

On the DHR end, deletions up to a single repeat were allowed. After the C-beta atoms are superimposed, a RMSD check across 9 residues was performed to ensure that the fusion results in a continuous helix. If no residues in the fused structure clash (Rosetta centroid energy < 10), sequence design was carried out at all positions within 8 Å of the junction. This first step of the fusion sampling is incorporated in the Rosetta MergePDBMover[15]. After additional sequence design around the junction region[25,26], fusions were then evaluated based on the number of helices interacting across the interface (at least 3), buried surface (sasa > 800) across the junction, and shape complementarity (sc > 0.6) to identify designs likely to be rigid across the junction point. In total, the building block library generated in silico by HF using HB hubs and DHR arms in this set consists of 490 C2s, 1255 C3s, 107 C5s, and 87 C6s.

As a proof of concept, select fusions to C5 (5H2LD-10)[10] and C6 (6H2LD-8)[27] HBs were tested experimentally, as structures of higher cyclic symmetries have been more difficult to design, resulting in a lack of available scaffolds, and larger structures are easier to experimentally characterize via electron microscopy. Out of 65 designs expressed in E.coli, 45 were soluble, 23 were monodisperse by SEC, and through SAXS analysis 7 had matching SAXS profiles compared to that from the computed design model (Supplementary Figs. 4–5). Cryo-electron microscopy of C5_HF-3921 followed by 3D reconstruction showed that the positions of the helical arms are close to the design model (Fig. 2e, Supplementary Figs. 8 and 9). Class averages of negative-stain electron microscopy (EM) of C5_HF-2101, C5_HF-0019, C6_HF-0075, and C6_HFuse-0080 (Fig. 2f–i respectively) clearly resemble predicted projection maps (Supplementary Figs. 10, 12, 13, and 14 respectively). Off-target states with fewer or greater numbers of DHR arms than the design models are also observed for C5_HF-0007 (Supplementary Fig. 11) and C6_HF-0075 (Supplementary Fig. 13), and to a lesser extent in C5_HF-0019 (Supplementary Fig. 12).

To explore the applicability of the HelixFuse method beyond de novo designed helical bundles, we also applied it to two non-helical bundle oligomers—1wa3, a native homo-trimer[28], and tpr1C4_pm3, a designed homo-tetramer[6]. We fused DHRs to the N-terminal helix of 1wa3 and the C-terminal helix of tpr1C4_pm3. For 1wa3, from the 13 designs that were expressed for experimental validation, 10 displayed soluble expression and had monodispersed peaks by SEC. We were able to solve the X-ray crystal structure of C3_nat_HF-0005 to 3.32 Å resolution (Fig. 2c). Sixteen tpr1C4_pm3 fusions were tested, 14 found to be soluble, and 10 displayed monodispersed peaks by SEC. C4_nat_HF-7900 was found to form monodisperse particles by cryo EM, with the 3D reconstruction modeled to 3.7 Å resolution (Fig. 2d, Supplementary Figs. 5–7). Both the crystal structure of C3_nat_HF-0005 and the model of the cryo-EM reconstruction of C4_nat_HF-7900 match the corresponding design models near the oligomeric hub of the protein where side chains are clearly resolved; but they deviate from the design model at the most distal portions of the structure, likely due to the inherent flexibility of the unsupported terminal helices of the DHRs[15,22,29] and lever arm effects which increase with increasing distance from the fusion site (Supplementary Fig. 15).

While the homo-oligomeric fusions are good building blocks for objects with higher-order point group symmetries, hetero-oligomeric fusions are needed for building more general classes of structures, and connecting different axes of symmetry in higher-order architectures (described below). To extend the complexity of structures that can be generated, we used the HelixFuse method to generate heteromeric two chain building blocks by fusing repeat proteins to two hetero-dimeric helical bundles (DHD-13, DHD-37)[20] (Fig. 1a). The fusion steps are identical, except for an

additional step of merging the chain A and chain B fusions and checking for clashes and incompatible residues. In total, 2740 heterodimers were generated in silico and added to the library.

With a sufficiently high design success rate, it is not essential that the building blocks be experimentally verified before being used to build larger structures. Since all building blocks terminate in repeat proteins that can be fused anywhere along their length, the total number of possible three building block fusions which can be built from this set is extremely large, which is important for finding solutions despite the degree of freedoms lost to symmetry constraints. The combined library consists of both HD and HF generated building blocks; overall, the HF structures tended to have smaller interfaces across the junction (for both methods, each junction contains ~15–30 mutations (see Supplementary information for sequence alignments)). While the HF are less globular than their HD counterparts, the smaller interface may contribute to the higher fraction of designs being soluble (~70% vs ~55%). The HD method also requires an additional step of building a new loop between the HB and DHR, which is another potential source of modeling error, and takes significantly more computational hours. Overall, the fraction with single dominant species in SEC traces (examples shown in Supplementary Figs. 1–5) profiles are similar (~35%).

**Higher order architectures with WORMS**. To generate a wide range of novel protein assemblies without interface design, we took advantage of the protein interfaces in the library of building blocks described in the previous section, which are oligomers with repeat protein arms. Assemblies are formed by splicing together alpha helices of the repeat protein arms in different building blocks. In our implementation, the user specifies a desired architecture and the symmetries and connectivity of the constituent building blocks. The method then iterates through splices of all pairs of building blocks at all pairs of (user specified, see "Methods") helical positions; this very large set is filtered on the fly based on the rms of the spliced helices, a clash check, off-architecture angle tolerance, residue contact counts around splice, helix contact count, and redundancy; all of which can be user-specified parameters (see "Methods"). The rigid body transform associated with each splice passing the above criteria is computed; for typical pairs of building blocks with 100 possible fusion sites, ($100 \times 100 = 10,000$) unfiltered splices are possible. With 100 choices of building blocks, the number of possible two-way splices is then ($100 \times 100 \times 10,000 = 10^8$).

This rapid symmetric architecture assembly through building block fusion has been implemented in a program called WORMS (Wm) which provides users with considerable control over building block sets, geometric tolerances, and other parameters and enables rapid generation of a wide range of macromolecular assemblies. With hundreds to thousands of building blocks each with potentially hundreds of residues available for fusion, the total number of three-way fusions is on the order of >$10^{14}$, so optimization of efficiency in both memory usage and CPU requirements was critical in WORMS software development, in particular hashing of the geometric transforms induced by fusion of building block fragments (see Methods). Once building block combinations are identified that generate the designed architecture (within a user specifiable tolerance), explicit atomic coordinates are calculated and used for clash checking, redundancy filtering, and any other filtering that requires atomic coordinates. Models for each assembly passing user-specified tolerances are constructed in Rosetta, scored and output for subsequent sequence design.

**Ring-shaped cyclic crown assemblies**. First, we explored the generation of C3, C4, and C5 assemblies with WORMS using

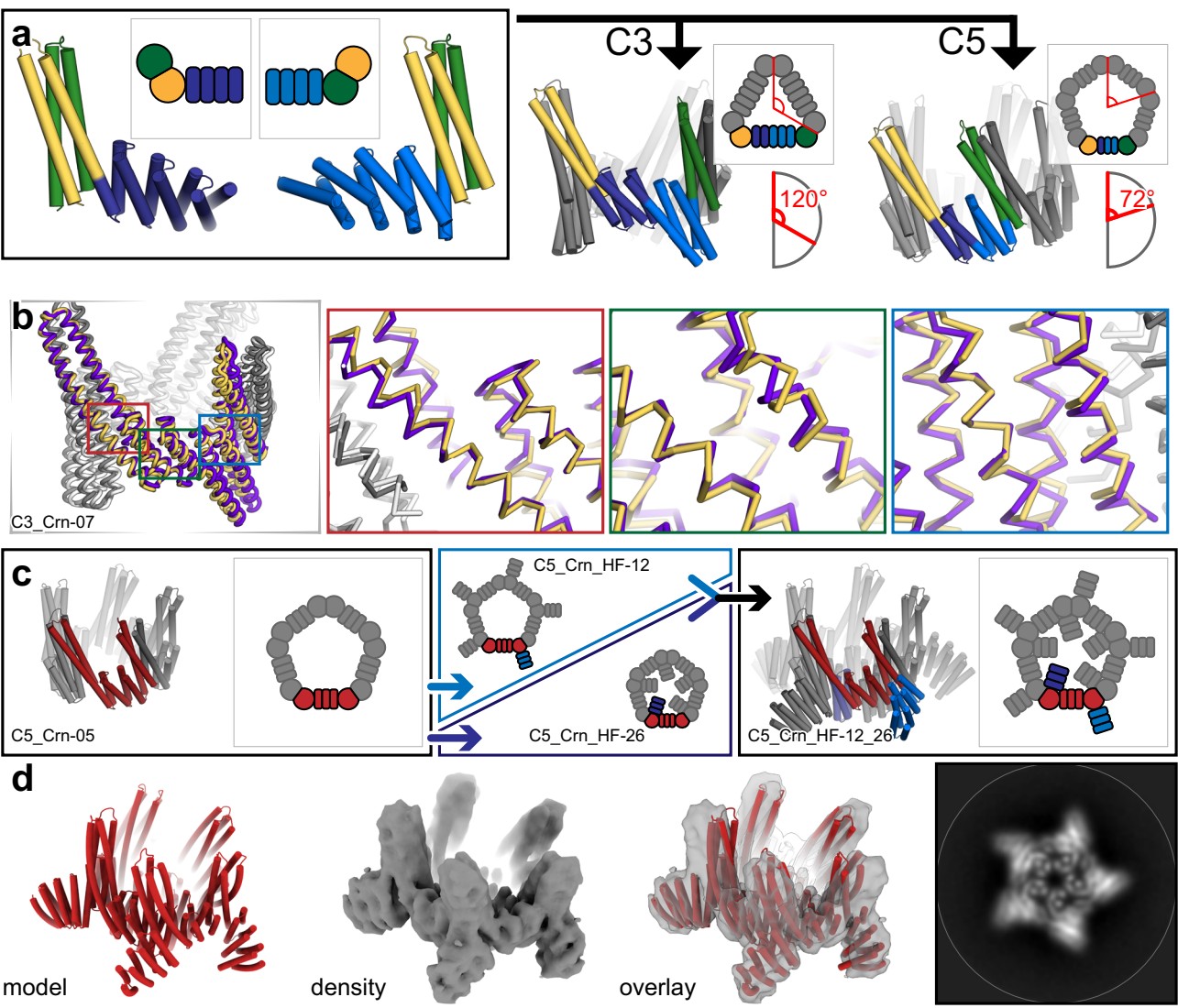

**Fig. 3 Design of cyclic crowns from heterodimeric building blocks. a** Hetero-dimeric HB (green/yellow) fused with different DHRs (shades of blue) were fused together using WORMS by enforcing a specific overall cyclic symmetry (C3 and C5 shown). **b** The backbones of the crystal structure (yellow/white) of C3_Crn-05 overlaid with the design model (purple/gray). Insets show the backbone matching focused at each of the fusion locations. **c** A C5 crown (C5_Crn-07, asymmetric unit in red) was fused to DHR units on either outward facing ("C5_Crn_HF-12", blue arrow) or inward facing termini ("C5_Crn_HF-26", dark blue arrow). The two structures were then merged together to generate a double fusion ("C5_Crn_HF-12_26", black arrow). **d** Cryo-EM class average of the fused 12_26 structure; the major C5 species shown. 3D reconstruction shows the main features of the designed structure are present, as is also evident in the class average (right).

heterodimer fusions. This resulted in head-to-tail cyclic ring structures dubbed "crowns" (Crn) (Fig. 3a).

Following fusion, the junction residues were redesigned to favor the fusion geometry and filtered as above. Seven C3s, seven C4s, and eight C5s were selected and tested experimentally. All yielded soluble protein, and 6, 2, and 1 respectively showed a single peak at the expected elution volume via SEC. We solved the structure of the C3_Crn-05 to 3.19 Å resolution (Fig. 3b). The overall topology is as designed and the backbone geometry at each of the three junctions is close to the design model. A deviation at the tip of the undesigned heterodimeric HB is likely to due to crystal packing. C5_Crn-07 chromatographed as a single peak by SEC and was found to be predominantly C5 by negative-stain EM (Fig. 3d), but minor off-target species (C4, C6, and C7) were also observed (Supplementary Fig. 16). Each of these structures experimentally verifies three distinct helical fusions (two HF, one WORMS) from the building block library that had not been characterized in isolation, confirming that the

building blocks in the *in silico* library can have sufficient accuracy for subsequent building efforts.

To further increase the diversity of the crown structures, we carried out a second round of HF on both termini of C5_Crn-07 (Fig. 3c). Six (6) N-terminal and 24 C-terminal fusions were selected and experimentally tested. All were soluble, but also had large void volume fractions when analyzed by SEC. When the peaks around the expected elution volumes were analyzed by negative-stain EM, ring-like structures were found in many of the samples. To facilitate EM structure determination, we combined a c-terminal fusion (C5_Crn_HF-12) and an n-terminal (C5_Crn_HF-26) fusion to generate C5_Crn_HF-12_26 (Fig. 3c), which resulted in a much cleaner and monodisperse SEC profile (Supplementary Fig. 17). Cryo-electron microscopy of 12_26 revealed the major population was C5 (77%) structures consistent with the design, in addition to C4 (1%), D5 (8%), and C6 (12%) subpopulations (Supplementary Fig. 17). We hypothesize that the D5 structure is due to transient interactions of histidines placed on

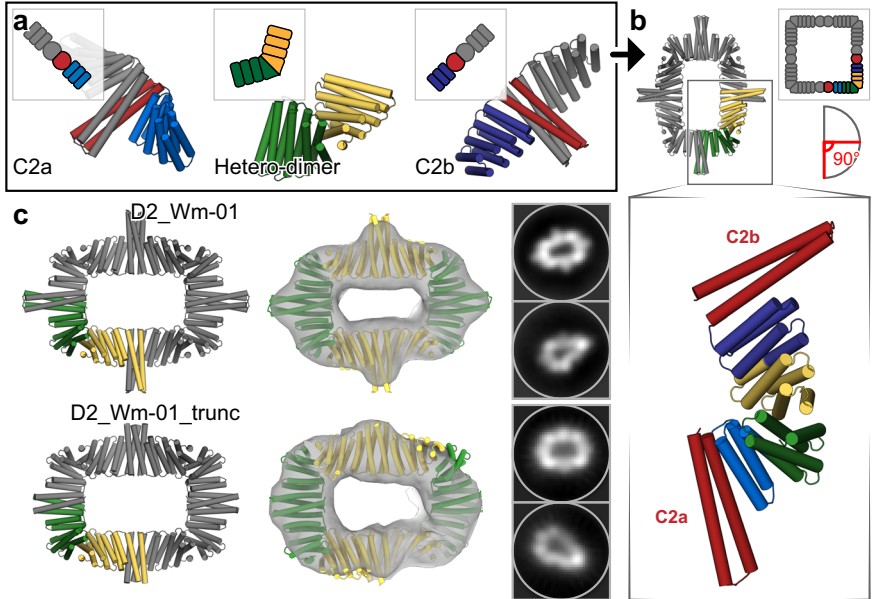

**Fig. 4 Design of two-component dihedral rings. a** Two different homodimeric HBs (red) with DHR extensions (shades of blue) were aligned to their respective symmetrical axes with dihedral symmetry. An additional heterodimer (green/yellow) was placed between them through architecture aware helical fusion, generating an 8-chain D2 ring. **b** The final asymmetric unit shown in green/yellow while the inset preserves the original colors. **c** Negative-stain EM followed by 2D average and 3D reconstruction of D2_Wm-01 and D2_Wm-01_trunc show that the major features of the designs were recapitulated (left) designed model, (middle) overlay of the designed models with the 3D reconstructions, (right) 2D averages.

the loops for protein purification. The final 3D reconstruction to 5.6 Å resolution shows that the major characteristics of the design model are present, despite some splaying of the undesigned portion of the heterodimeric HB relative to the design model (Fig. 3d).

**Dihedral assemblies with perpendicular helical bundles.** Dihedral protein assemblies are attractive building blocks for making higher-order 2D arrays and 3D crystal protein assemblies, and can be useful for receptor clustering in cellular engineering[30]. We first set out to design dihedral protein assemblies of D2 symmetry. C2 homo-oligomers with DHR termini (described above) were fused with *de novo* hetero-dimers using WORMS (schematics shown in Fig. 4a, b); the symmetry positions the two C2 axes perpendicular to one another. The D2 rings contain in total of eight protein chains, with two chains (two-component) as the asymmetric unit. To generate these rings, we used a database of building blocks containing seven homo-dimers and one heterodimer.

Of 208 outputs, we selected six designs to test, out of which three expressed as soluble two-component protein assemblies as indicated by Ni-NTA pulldown and subsequent SDS-PAGE experiments. Of these, two designs (designated as D2_Wm-01 and D2_Wm-02) eluted as expected by SEC and had SAXS profiles that matched with the designed models (Supplementary Figs. 18 and 19).

To characterize the structures of D2_Wm-01 and D2_Wm-02 in more detail, we performed negative-stain EM and subsequent 2D averaging and 3D refinement. 2D averaging and ~16 Å resolution 3D election density maps are consistent with the design model (Fig. 4c, Supplementary Fig. 19). The homo dimeric building blocks used in D2_Wm-01 and D2_Wm-02 have large interface areas (~35 residues long; 5 heptads). We sought to reduce the interface area by truncating the helices to facilitate expression of the components and reduce off target interactions. Deletion of one heptad from either of the homodimers of D2_Wm-01 (designated D2_Wm-01_trunc) resulted in a single and much narrower SEC peak of the expected molecular weight (Supplementary Fig. 18). Negative-stain EM followed by 2D

averaging and 3D refinement indicated monodispersed particles with structure matching the designed model (Fig. 4c).

**Point group symmetric assemblies.** We next sought to generate tetrahedral nanocages by fusing C3 to C2 building blocks through repeat protein monomers such that the C3 and C2 axes align with those on tetrahedral symmetry. A C3 HB[19] and idealized ankyrin homo-dimers[6] were connected through a monomeric ankyrin scaffold to generate nanocages capable of hosting engineered DARPIN binding sites (Fig. 5a).

In total, 27 valid fusion combinations generating the tetrahedral architecture were identified, of which 20 involved ankyrin homo-dimer extension at the N-terminus and the remaining 7 at the C-terminus. Following sequence design, 8 selected constructs were expressed and two were found to be soluble with mono-disperse elution profile peaks by SEC. These contain different C3 HBs identical backbone geometry, but with different internal hydrogen-bond networks. One (T_Wm-1606) was selected for negative-stain EM and discrete particles were observed whose 2D class averages and 20 Å 3D reconstruction matched the computational model (Fig. 5b). There was also good agreement between experimental SAXS profiles and profiles computed from the design model (Supplementary Fig. 20).

Icosahedral symmetry nanocages have been previously designed using docking followed by interface design[8–10]. To build such structures by helix fusion using our building blocks with smaller and weaker interfaces that give rise to cooperative assembly[31–33], we systematically split each DHR at the loop in the center of four repeats, resulting in a hetero-dimeric structure with two repeats on each side. The resulting interfaces are considerably smaller than in, for example, our *de novo* designed HBs. Tetrahedral, octahedral, and icosahedral assemblies were generated from C5, C3, and C2 building blocks placed on their respective point group symmetry axes, connected by the split DHRs heterodimers (Fig. 5c). Following fusion, sequence design was performed at each of the two new junctions.

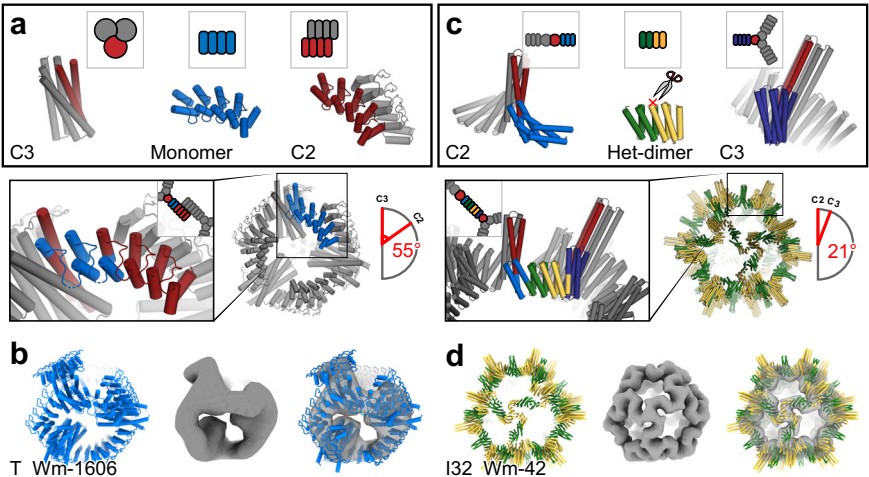

**Fig. 5 Design of assemblies with point group symmetry. a** Tetrahedron design schematic. A HB and a C2 homo-oligomeric made from ankyrin repeat proteins were aligned to their respective tetrahedral symmetry axis (red), and connected via fusion to Ankyrin repeat monomers (blue) to generate the target architecture. **b** 3D reconstruction reveals a well fitting map of T_Wm-1606. **c** Icosahedral design schematic. Libraries of unverified cyclic fusion homo-dimers and trimers were aligned to the corresponding icosahedral symmetry axes. Using WORMS, fusions to DHRs split in the center that hold the two homo-oligomers in the orientations which generate icosahedral structures were identified. **d** Cryo-EM 3D reconstruction of I32_Wm-42 closely matches the designed model.

Fifty-seven total designs were selected for experimental characterization; 25 co-eluted by Ni-NTA chromatography, and of these seven designs had large peaks in the void volume in SEC chromatography as expected for particles of this size. When the peaks were collected and re-analyzed with a Sephacryl 500 column, one design, I32_Wm-42 (icosahedral architecture) was resolved into a void and a resolved peak (Supplementary Fig. 21). Cryo-EM analysis of the resolved peak reveals well-formed particles that when reconstructed to 9 Å resolution, closely match the design model, including the distinct "S" shaped turn between the C3 and C2 axes (Fig. 5d). This structure is considerably more open than previous icosahedral cages built by designing non-covalent interfaces between homo-oligomers. For another design, T32_Wm-24, while the full cage did not assemble, we were able to crystallize the polar-capped trimer component (C3_HF_Wm-0024A) and solve the structure by X-ray diffraction to 2.69 Å (Fig. 2b). The structure clearly shows that both of the newly designed junctions (from HF and WORMS) are as designed, matching the design model.

The 120 subunit I32_Wm-42 icosahedral nanocage has a molecular weight of 3.4 MDa and a diameter of 42.7 nm and illustrates the power of our combined hierarchical approach. I32_Wm-42 is constructed from five building blocks (two HBs and three repeat proteins) combined via four unique rigid junctions (with on average 25 amino acid substitutions each); the EM structure demonstrates that all were modeled with reasonable accuracy. The combination of the HD and HF helix fusion methods created a large set of over 1500 oligomeric building blocks from which WORMS was able to identify combinations and fusion points that generated the icosahedral architecture; this example is notable because none of the fused oligomeric building blocks had been previously characterized experimentally. With fewer unknowns, either using less segments or a larger fraction of previously validated building blocks, we expect considerable improvement of the overall success rate.

## Discussion

Our general rigid helix-fusion-based pipeline fulfills the promise of early proposals[14,34] in providing a robust and accurate procedure for generating large protein assemblies by fusing symmetric building blocks and avoiding interface design, and should streamline assembly design for applications in vaccine development, drug delivery, and biomaterials more generally. The set of structures generated here goes considerably beyond previous work with rigid helical fusions[14,16,35], and the "WORMS" software introduced here is quite general and readily configurable to different nanomaterial design challenges. WORMS can be easily extended to other symmetric assemblies including 2D arrays and 3D crystals, and should be broadly useful for generating a wide range of protein assemblies.

In recent years, a wide range of macromolecular assemblies have been created by using metal-mediated interactions[36,37], designed non-covalent interactions[8–10,38], and fusion approaches[14,16,35]. These materials have a wide range of remarkable structures and material properties. However, they have been generally limited to the building blocks provided by natural protein evolution, and other than the fusion approach, require engineering of protein interfaces which can differ from case to case (metal-mediated interactions have more transferability). Our methodology extends fusion-based approaches in two ways, first by developing systematic procedures for creating large sets of *de novo* building blocks, and second, by developing efficient software for scanning through the exponentially large sets of possible multi-way fusions of these building blocks for those that generate any desired overall architecture.

DNA nanotechnology has had advantages in modularity and simplicity over protein design because the basic interactions (Watson-Crick base pairing) and local structures (the double helix) are always the same. Proteins in nature exhibit vast diversity compared to duplex DNA, and correspondingly, re-engineering naturally occurring proteins and designing new ones has been a more complex task than designing new DNA structures. The large libraries of "clickable" building blocks—helical bundle—repeat protein fusions—and the generalized WORMS software for assembling these into a wide range of user specifiable architectures that we present in this paper are a step towards achieving the modularity and simplicity of DNA nanotechnology with protein building blocks. Although this modularity comes at some cost in that the building blocks are less diverse than proteins

in general, they can be readily functionalized by fusion to protein domains with a wide range of functions. We show that it is possible to genetically fuse DHR "adapters" to natural proteins; these proteins can then be used in larger assemblies through WORMS with less likelihood of disrupting the original protein fold. Proteins of biological and medical relevance (binders like protein A, enzymes, etc.) can be used as components and combined with *de novo* designed HBs and DHRs to form nanocages and other architectures.

Moving forward, there are still a variety of challenges to address. The larger the set of building blocks for WORMS the more precisely the geometric constraints associated with the desired architecture can be achieved, and hence it is advantageous to use the very large *in silico* libraries of building blocks that can be created by the HB— repeat protein fusion rather than the very much smaller sets of fusions that can be experimentally characterized in advance (tens of thousands compared to tens). It will be important to understand how uncertainties in the structures of the *in silico* fusions translate into uncertainties in the structures of the resulting architectures, and more generally, how to further improve the fusion approach so that the *in silico* structures are nearly perfectly realized. As the assemblies become more complex with different building blocks and total number of subunits, more alternative structures become possible. Understanding how to achieve cooperative assembly and controlling for the specificity of the desired assembly over alternatives will be an increasingly important challenge as the complexity of the target nanomaterials increases.

## Methods

**RosettaRemodel forward folding.** To test the extent to which the designed sequences encode the designed structure around the junction site, we used large-scale *de novo* folding calculations. Due to computational limitations with standard full chain forward folding[39,40], we developed a similar but alternate approach for larger symmetric structures. Using RosettaRemodel[24] in symmetry mode (reversing the anchor residue for cases where the HB was at the C-terminus), we locked all residues outside the junction region as rigid bodies, only allowing 40 residues starting from the end of the HB in the primary sequence direction of the DHR to be re-sampled. The blueprint file was set up to be agnostic of secondary structure in this segment of protein and we deleted all DHR residues past the first two helices after the rigid body region to reduce CPU cost. Each structure was set to at least 2000 trajectories to create a forward folding funnel.

**WORMS.** The WORMS software overall requires two inputs, a database of building block entries (format described in Supplementary Information in detail) and a configuration file (or command line options) as described in the main text to govern the overall architecture. While some segments can be of single building blocks of interest, to generate a wide variety of outputs, tens to thousands of entries per segment should be used. The number of designs generated also depends on the number of fusion points allowed, as the size of the space being sampled increases multiplicatively with the number of segments being fused. There are many options available to the user to control the fusions which are output as solutions; we have tuned the default options to be relatively general-use (see Supplementary Information for a description of options). A key parameter is the *tolerance*, he allowed deviation of the final segment in the final structure away from its target position given the architecture. For different geometries the optimal values vary; for example, the same tolerance values involve more drastic error in icosahedral symmetry than cyclic symmetry. The WORMS code is specifically designed to generate fusions that have a protein core around the fusion joint; unless specified using the *ncontact_cut*, *ncontact_no_helix_cut*, and *nhelix_contacted_cut* option set, the code will not produce single extended helix fusions.

**Gene preparation.** All amino acid sequences derived from Rosetta were reverse translated to DNA sequences and placed in the pET29b+ vector. For two-component designs, all designs were initially constructed for bi-cistronic expression by appending an additional ribosome binding site (RBS) in front of the second sequence with only one of the components containing a 6xHis tag. Genes were synthesized by commercial companies: Integrated DNA Technologies (IDT), GenScript, Twist Bioscience, or Gen9.

**Protein expression and purification.** All genes were cloned into *E. coli* cells (BL21 Lemo21 (DE3)) for expression, using auto-induction[41] at 18° or 37 °C for 16–24 h in 500 mL scale. Post-induction, cultures were centrifuged at 8000 × *g* for 15 min. Cell pellets were then resuspended in 25–30 mL lysis buffer (TBS, 25 mM Tris, 300 mM NaCl, pH8.0, 30 mM imidazole, 0.25 mg/mL DNase I) and sonicated for 2 min total on time at 100% power (10 s on/off) (QSonica). Lysate was then centrifuged at 14,000 × *g* for 30 min. Clarified lysates were filtered with a 0.7 µm syringe filter and put over 1–4 mL of Ni-NTA resin (QIAgen), washed with wash buffer (TBS, 25 mM Tris, 300 mM NaCl, pH8.0, 60 mM imidazole), then eluted with elution buffer (TBS, 25 mM Tris, 300 mM NaCl, pH8.0, 300 mM imidazole). Eluate was then concentrated with a 10,000 m/w cutoff spin concentrator (Millipore) to ~0.5 mL based on yield for SEC.

D2 proteins went through an extra round of bulk purification. Concentrated protein was heated at 90 °C for 30 min to further separate bacterial contaminants. Samples were then allowed to cool down to room temperature and any denatured contaminants were removed by centrifuging at 20,000 × *g*.

**Size exclusion chromatography (SEC).** All small oligomers were passed through a Superdex200 Increase 10/300 GL column (Cytiva) while larger assemblies were passed through a Superose 6 Increase 10/300 GL column (Cytiva) on a AKTA PURE FPLC system. The mobile phase was TBS (TBS, 25 mM Tris, 300 mM NaCl). In addition, for the icosahedral assembly, an additional custom packed 10/300 Sephacryl500 column (Cytiva) was used to separate out the void. Samples were run at a speed of 0.75 mL/min and eluted with 0.5 mL fractions.

**Protein characterization.** See supplementary information for detailed methods regarding SAXS sample preparation, electron microscopy, and x-ray crystallography.

## Data availability

Crystallography data. C3_HD-1069 (6XT4). C3_HF_Wm-0024A (6XI6). C3_nat_HF-0005 (6XH5). C3_Crn-05 (6XNS). C4_nat_HF-7900 (6XSS). Electron microscopy data. C4_nat_HF-7900 (EMD-22305). C5_HF_3921 (EMD-22306). C5_Crn_HF-12_26 (EMD-23173). (EMD-23174). D2_Wm-01. (EMD-23168). D2_Wm-01_trunc (EMD-23170). D2_Wm-02 (EMD-23169). T_Wm-1606 (EMD-23171). I32_Wm-42 (EMD-23172). Source data are provided with this paper.

## Code availability

Full source code and repository for the WORMS software can be found on GitHub: https://github.com/willsheffler/worms; https://doi.org/10.5281/zenodo.4323517. The Rosetta software suite is available for free to all non-commercial users and to commercial users for a fee: https://www.rosettacommons.org/. Additional Rosetta related scripts and files can be found as source data.

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

## Acknowledgements

This work was supported by the National Science Foundation (NSF) award 1629214 (DB), a generous gift from the Audacious Project, the Open Philanthropy Project Improving Protein Design Fund, the National Institute of General Medical Sciences (R01GM120553 to D.V.), the National Institute of Allergy and Infectious Diseases (HHSN272201700059C to D.V.), a Pew Biomedical Scholars Award (D.V.), an Investigators in the Pathogenesis of Infectious Disease Award from the Burroughs Wellcome Fund (D.V.) and the University of Washington Arnold and Mabel Beckman cryo-EM center. Y.H. was supported in part by a NIH Molecular Biology Training Grant (T32GM008268). R.M. is a recipient of the Washington Research Foundation (WRF) Innovation fellowship and his research is funded in part by the US DOE BES Energy Frontier Research Center CSSAS (The Center for the Science of Synthesis Across Scales) located at the University of Washington (award number DESC0019288). U.N. was supported in part by PHS NRSA (T32GM007270) from NIGMS. A.C. is a recipient of the Human Frontiers Science Program Long Term Fellowship and a Washington Research Foundation Senior Fellow. This work was conducted at the Advanced Light Source (ALS), a national user facility operated by Lawrence Berkeley National Laboratory on behalf of the Department of Energy, Office of Basic Energy Sciences, through the Integrated Diffraction Analysis Technologies (IDAT) program, supported by DOE Office of Biological and Environmental Research. Additional support comes from the National Institute of Health project ALS-ENABLE (P30 GM124169) and a High-End Instrumentation Grant S10OD018483. We thank staff at Advanced Photon Source beamline NE-CAT 24-ID-E for data collection. Northeastern Collaborative Access Team beamline supported by NIH grants P30GM124165 and S10OD021527, and DOE contract DE-AC02-06CH11357. We also want to thank Banumathi Sankaran at the Advanced Light Source (ALS) beamline 8.2.2 at Lawrence Berkeley National Laboratory for data collection. The Berkeley Center for Structural Biology is supported in part by the National Institutes of Health (NIH), National Institute of General Medical Sciences, and the Howard Hughes Medical Institute. The Advanced Light Source (ALS) is supported by the Director, Office of Science, Office of Basic Energy Sciences and US Department of Energy under contract number DE-AC02-05CH11231. We thank Kristen Dancel-Manning and Alice Liang of the NYU Microscopy Laboratory, William Rice and Bing Wang of the NYU Cryo-EM Laboratory, Kashyap Maruthi, Ed Eng, Laura Yen, and Misha Kopylov of the New York Structural Biology Center, and members of the Bhabha/Ekiert labs for assistance with grid screening and data collection and helpful discussions. We especially thank Nicolas Coudray of the Bhabha/Ekiert lab for helpful discussions and guidance regarding EM data processing. Some of this work was performed at the Simons Electron Microscopy Center and National Resource for Automated Molecular Microscopy located at the New York Structural Biology Center, supported by grants from the Simons Foundation (SF349247), NYSTAR, and the NIH National Institute of General Medical Sciences (GM103310) with additional support from Agouron Institute (F00316), NIH (OD019994), and NIH (RR029300). We would like to thank the Rosetta@Home user base for donating their computational hours to run our forward folding simulations. Thanks to George Ueda for the unpublished tj18_asym13 heterodimer. An additional thanks to Robby Divine and Josh Lubner for support in the documentation and development WORMS.

## Author contributions

Y.H., R.M., and D.B. wrote the manuscript. Y.H., W.S., and T.B. developed the HelixDock protocol; Y.H., R.M., N.I.E., I.V., and U.N. made designs and characterized experimentally with assistance from E.T., A.S., and C.M.C. in protein production. Y.H. and T.B. developed the HelixFuse protocol. I.V. developed the helical fusion method in.NET; W.S. and D.B. implemented it into the WORMS protocol; Y.H. and R.M. assisted developing in its application. U.N. and E.T. crystallized and M.J.B. solved the C3_HD-1069 structure. A.K. crystallized C3_nat_HF-0005, C3_HF_Wm-0024A, and C3_Crn-05. A.B. solved the crystal structure for C3_nat_HF-0005 and C3_HF_Wm-0024A. M.J.B. solved the structure for C3_Crn-05. R.L.R., assisted by D.E. and G.B., performed negative-stain and cryo-EM for all HelixFuse structures presented. Y.H. designed and characterized crown structures and icosahedral cage; R.M. the dihedral structures, I.V. the tetrahedral cage. R.M. and A.C. performed initial EM screening of dihedral, cyclic and icosahedral WORMS structures. Y.J.P., assisted by D.V., performed negative-stain and cryo-EM for all WORMS structures presented. D.B. guided the project.

## Competing interests

The authors declare no competing interests.
