## [Peer Review File · Nature Communications]

***The previous round of reviews was undertaken with another Nature journal.**

REVIEWERS' COMMENTS

Reviewer #2 (Remarks to the Author):

The authors have addressed my minor comments on points of clarity. Another referee points out that the methodology is not all that new. However the importance and difficulty of the problem and the sheer scope of the applications carried out make the paper completely suitable for Nature Communications imo (or Nature Chemistry for that matter).

Reviewer #3 (Remarks to the Author):

I think the revised paper is ready for publication. It would need a careful read through - I doubt the authors meant to leave "Tezcan reference here" (p. 3, line 55) for the submission - but I don't have any further comments.

REVIEWERS' COMMENTS

Reviewer #2 (Remarks to the Author):

The authors have addressed my minor comments on points of clarity. Another referee points out that the methodology is not all that new. However the importance and difficulty of the problem and the sheer scope of the applications carried out make the paper completely suitable for Nature Communications imo (or Nature Chemistry for that matter).

We thank the reviewer for the comments and recommendation.

Reviewer #3 (Remarks to the Author):

I think the revised paper is ready for publication. It would need a careful read through - I doubt the authors meant to leave "Tezcan reference here" (p. 3, line 55) for the submission - but I don't have any further comments.

We have edited the remaining reference text into its appropriate format.